# MV-Diffus3R: Refining Multi-View Diffusions for Geometric Coherence 3D Reconstruction

**MV-Diffus3R Pipeline**

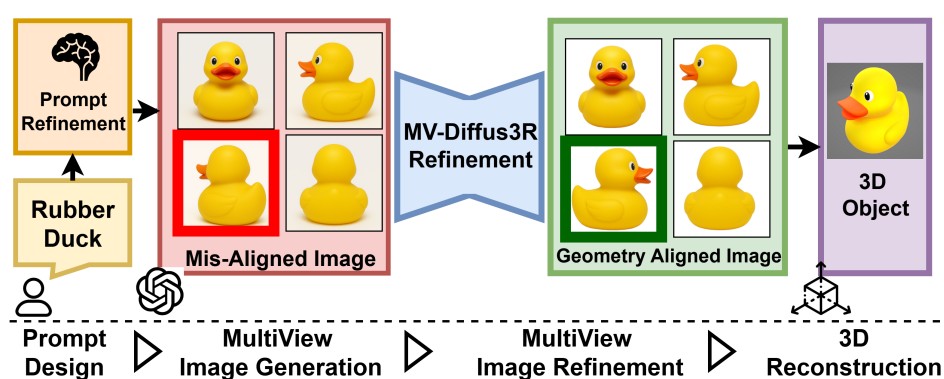

Figure 1: **MV-Diffus3R Pipeline Overview.** Our two-stage approach decouples view generation from geometric refinement. GPT-generated multi-view images often exhibit geometric inconsistencies such as incorrect rotational angles (bottom-left example), which compromise 3D reconstruction quality. MV-Diffus3R serves as a plug-and-play refinement module that transforms inconsistent multi-view sets into geometrically coherent representations suitable for high-quality 3D reconstruction.

## ABSTRACT

Recent breakthrough text-to-image models like GPT achieve unprecedented photorealistic quality, yet our analysis reveals critical geometric inconsistencies when leveraging these models for multi-view generation. These inconsistencies manifest as specific rotational errors—such as facial expressions changing between views (open mouth becoming closed) or object details disappearing during rotation (remote control buttons missing in side views)—alongside systematic texture loss that compromises downstream 3D reconstruction quality. While existing methods attempt to address multi-view consistency through end-to-end generation with geometric constraints, they face an inherent trade-off between visual fidelity and geometric coherence, often producing over-smoothed results that sacrifice the exceptional detail quality achievable by models like GPT. To harness the full potential of these powerful 2D foundation models while resolving their geometric limitations, we introduce a novel two-stage pipeline that strategically decouples view generation from geometric refinement. Our core contribution is MV-Diffus3R, a specialized plug-and-play refinement module that takes high-quality but geometrically inconsistent multi-view images from GPT and produces geometrically coherent outputs suitable for high-quality 3D reconstruction. MV-Diffus3R employs Plücker ray embeddings for precise geometric conditioning and a dual-pathway attention mechanism that simultaneously preserves fine visual details while enforcing cross-view geometric correspondence. Through comprehensive evaluation on GPT-generated multi-view sets, we demonstrate superior geometric fidelity compared to existing methods, achieving 33% FID improvements while maintaining exceptional visual quality.

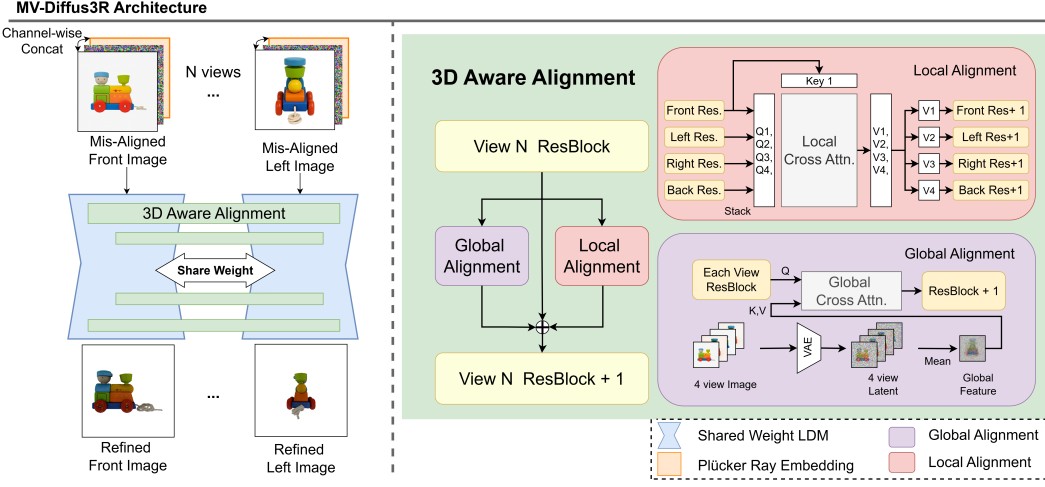

Figure 2: **MV-Diffus3R refinement module.** Given four geometrically misaligned input views with detail loss from text-to-image generators, the model leverages Plücker ray embeddings as geometric conditioning and applies 3D-aware alignment modules to produce geometrically coherent outputs while preserving high-frequency visual details.

# 1 INTRODUCTION

State-of-the-art text-to-image models have achieved unprecedented photorealistic quality, yet critical geometric inconsistencies emerge when leveraging these models for multi-view generation. Our experimental analysis reveals that these inconsistencies manifest as rotational errors—facial expressions changing between views or object details disappearing during rotation—alongside systematic texture loss that fundamentally compromises downstream 3D reconstruction quality.

Current approaches face significant limitations in harnessing the capabilities of these powerful 2D foundation models. End-to-end text-to-3D methods such as DreamFusion Poole et al. (2022) and Magic3D Lin et al. (2023) produce over-smoothed results due to inherent tension between maintaining 3D consistency and preserving exceptional detail quality. Multi-view generation methods like Zero-1-to-3++ Shi et al. (2023), SyncDreamer Liu et al. (2024b), and MVDream Shi et al. (2024a) face an inescapable trade-off between visual fidelity and geometric coherence. Most critically, these monolithic approaches cannot exploit the full expressive power of foundation models without extensive modifications that compromise their exceptional qualities.

To overcome these limitations, we introduce a novel two-stage pipeline that strategically decouples view generation from geometric refinement. This architecture enables unrestricted utilization of existing 2D foundation models while dedicating a specialized refinement stage to correcting geometric inconsistencies. Our pipeline leverages these models without modification, followed by MV-Diffus3R (**M**ulti**V**iew **Diffus**ion for **3**D-aware **R**efinement), a plug-and-play module that transforms geometrically inconsistent multi-view sets into coherent representations suitable for high-quality 3D reconstruction.

MV-Diffus3R employs Plücker ray embeddings for precise geometric conditioning and a dual-pathway attention mechanism that preserves fine visual details while enforcing cross-view geometric coherence. The method operates without requiring camera pose estimation or 3D supervision, making it practically applicable to real-world generation workflows.

The main contributions of this work are as follows:

- A novel two-stage pipeline that decouples view generation from geometric refinement, enabling unmodified use of powerful 2D foundation models while achieving superior geometric consistency for 3D reconstruction

- MV-Diffus3R, a plug-and-play refinement module that corrects geometric inconsistencies and texture degradation in foundation model multi-view outputs, effectively bridging 2D visual quality and 3D structural requirements

- Comprehensive experimental validation demonstrating 33% FID improvement over existing methods while preserving exceptional visual quality and establishing efficiency suitable for practical deployment

## 2 RELATED WORK

### 2.1 TEXT-TO-3D GENERATION USING 2D PRIORS

Recent advances exploit strong 2D foundation models to bypass the lack of large-scale 3D data, primarily via Score Distillation Sampling (SDS) Poole et al. (2022). Early instantiations like Dream-Fusion Poole et al. (2022) and Magic3D Lin et al. (2023) demonstrated text-to-3D synthesis, with follow-ups improving fidelity and optimization: HiFA Zhu et al. (2024) (dual-space distillation and timestep annealing), ProlificDreamer Wang et al. (2023) (Variational Score Distillation), and preference-based tuning such as DreamDPO Zhou et al. (2025). To reduce optimization cost, amortized or feed-forward paradigms have been proposed, including sparse-view reconstruction and pseudo-image diffusion (e.g., Instant3D Li et al. (2023a), PI3D Liu et al. (2024a)) and student-teacher distillation schemes (e.g., ET3D Lorraine et al. (2023), GANFusion Attaiki et al. (2024)).

While existing methods produce visually plausible 3D content, they frequently suffer from geometric inconsistencies and misaligned views when considered jointly. Our approach addresses this limitation through specialized post-hoc refinement rather than end-to-end generation.

### 2.2 MULTI-VIEW CONSISTENCY IN GENERATIVE MODELS

Maintaining geometric coherence across views has been tackled in 3D-aware generative models, from early neural rendering GANs (e.g., HoloGAN Nguyen-Phuoc et al. (2019), GRAF Schwarz et al. (2021)) to more recent tri-plane and imitation-based designs like EG3D Chan et al. (2022) and Mimic3D Chen et al. (2023). A core difficulty stems from conflicting 2D priors leading to multi-front or canonical-view collapse Jain et al. (2022); Armandpour et al. (2023); remedies include prior fine-tuning and modified sampling Seo et al. (2024); Huang et al. (2024). Multi-view diffusion approaches (e.g., MVDream Shi et al. (2024b), Zero123++ Shi et al. (2023), SyncDreamer Liu et al. (2024b), MVDiffusion Tang et al. (2023), Era3D Li et al. (2024)) advance consistency-aware generation, but they still navigate a fidelity-vs-coherence trade-off. Crucially, none directly target post-hoc geometric refinement of small inconsistent view sets, which is the gap our method fills.

### 2.3 DIFFUSION-BASED IMAGE EDITING AND CONTROL

Controllable diffusion models have enabled sophisticated image editing and conditioning. Instruction-driven editing (InstructPix2Pix Brooks et al. (2023)), spatial conditioning modules (ControlNet Zhang et al. (2023), T2I-Adapter Mou et al. (2023b)), and style/attribute manipulation techniques (e.g., StyleDiffusion Li et al. (2023c), DiffusionCLIP Kim et al. (2022)) provide the backbone for targeted transformations. Inpainting and compositional edits have matured via methods like RePaint Lugmayr et al. (2022), Palette Saharia et al. (2022), and Paint by Example Yang et al. (2022), while recent control advances (DragDiffusion Shi et al. (2024c), DragonDiffusion Mou et al. (2023a), Delta Denoising Score Hertz et al. (2023), Contrastive Denoising Score Nam et al. (2024)) offer finer latent manipulation.

Our work extends this line with geometry-aware conditioning tailored for multi-view alignment: unlike general spatial control, our Plücker ray embeddings and dual-pathway attention explicitly encode 3D geometric relationships, enabling refinement that existing diffusion-control frameworks do not address.

## 3 PROPOSED METHOD

In this section, we present **MV-Diffus3R**, a multi-view to multi-view geometry enhancement module designed to address the geometric distortions and detail inconsistencies commonly produced by large generative models. The architecture leverages a dual-path alignment strategy that combines global geometry preservation with local view-dependent refinement to transform distorted multi-view image sets into geometrically coherent representations. Building upon the InstructPix2Pix framework, **MV-Diffus3R** incorporates specialized 3D-aware alignment modules and Plücker ray embeddings to maintain spatial consistency across viewpoints while correcting artifacts inherent in upstream generation processes. The model takes as input a set of distorted multi-view images along with orientation hints, and produces refined outputs suitable for downstream 3D reconstruction applications.

### 3.1 DATASET CONSTRUCTION

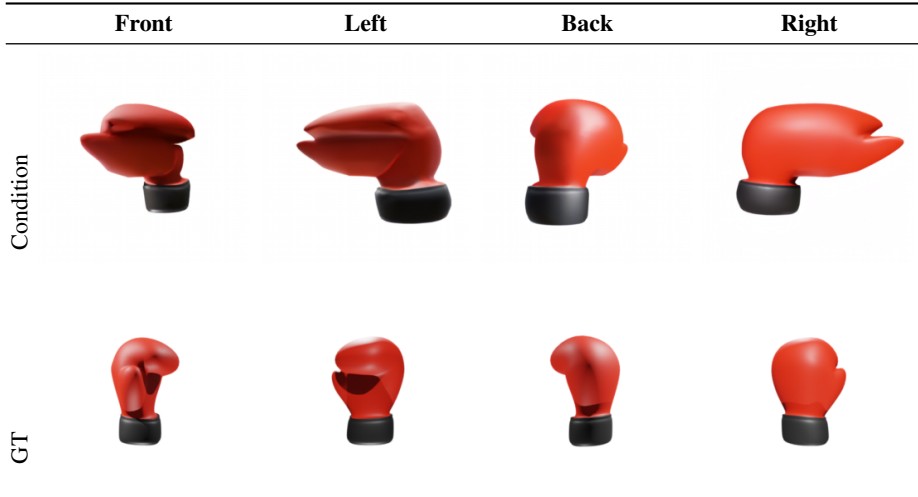

Figure 3: **Dataset visualization showing paired training examples.** Top row: SV3D-generated distorted views with geometric inconsistencies. Bottom row: corresponding Objaverse XL ground truth renderings across four orthogonal viewpoints.

To validate our decoupled pipeline architecture and specialized geometric refinement capabilities, we construct paired datasets that systematically capture the geometric inconsistencies characteristic of state-of-the-art generative models. Our dual-dataset approach directly supports the core contributions outlined in the introduction by providing controlled training scenarios and realistic evaluation conditions that demonstrate the effectiveness of separating view generation from geometric refinement.

Our training dataset leverages the Objaverse XL collection combined with SV3D Voleti et al. (2024)-generated distortions to create controlled geometric inconsistency patterns. For each selected object, we render 21 geometrically consistent ground truth views at distinct azimuthal angles with fixed elevation, then employ SV3D Voleti et al. (2024) to generate corresponding distorted multi-view sets from a single frontal input. This process systematically introduces the characteristic artifacts observed in single-view-to-multi-view generation, including asymmetrical features, shape deformations, and cross-view detail inconsistencies. We implement DINO similarity filtering with scores in the range [0.7, 0.9] to retain moderate distortions while excluding extreme geometric failures, yielding approximately 50,000 objects corresponding to 1.05 million total training images.

For evaluation, we construct a secondary dataset using the Google Scanned Objects Downs et al. (2022) collection paired with latest ChatGPT image generation model generated multi-view sets. This evaluation approach directly captures the geometric inconsistencies and viewpoint ambiguities encountered when using state-of-the-art text-to-image models for multi-view generation, particularly the challenges in maintaining left-right consistency and preventing axis confusion during text-

based rotation commands (complete prompt engineering specifications and rendering configurations provided in supplementary material). The resulting dataset provides realistic assessment conditions that validate our method's practical applicability to current generative model outputs, demonstrating the plug-and-play refinement capabilities central to our pipeline architecture.

## 3.2 MV-DIFFUS3R

MV-Diffus3R represents a specialized plug-and-play refinement module that addresses the previously unexplored task of multi-view geometric alignment without requiring additional camera pose estimation or 3D supervision. Given a set of four orthogonal views exhibiting geometric distortions produced by upstream generators, MV-Diffus3R leverages Plücker ray embeddings as geometric conditioning to produce refined, geometrically coherent multi-view outputs. This module serves as the geometric consistency engine in our decoupled pipeline architecture, enabling full utilization of existing 2D foundation models without modification while achieving superior geometric consistency.

### 3.2.1 ARCHITECTURE DESIGN

The architecture of MV-Diffus3R addresses the fundamental challenge of generating geometrically consistent multi-view image sets while preserving high-frequency visual details. Building upon the InstructPix2Pix framework for established image editing capabilities, we introduce our core innovation: a novel **3D-aware alignment module** that operates through a dual-pathway attention mechanism specifically designed for multi-view geometric refinement.

**Geometric Conditioning Through Plücker Ray Embeddings**   Our method employs Plücker ray embeddingsPlücker (1828) as the primary geometric conditioning mechanism, providing unambiguous spatial information to distinguish between visually similar but geometrically distinct viewpoints. As established in the preliminary section, these six-dimensional embeddings uniquely identify each ray in 3D space, enabling robust geometric disambiguation even when different camera poses produce visually similar projection patterns.

We extend the standard UNet input from 8 channels to 14 channels to accommodate this geometric conditioning. The concatenated input tensor is formulated as:

$$\mathbf{x}_{input} = \text{concat}(\mathbf{z}_t, \mathbf{c}_{img}, \mathbf{c}_{ray}) \in \mathbb{R}^{14 \times H \times W} \tag{1}$$

where $\mathbf{z}_t \in \mathbb{R}^{4 \times H \times W}$ represents the noised latent features at timestep $t$, $\mathbf{c}_{img} \in \mathbb{R}^{4 \times H \times W}$ contains the VAE-encoded conditioning images, and $\mathbf{c}_{ray} \in \mathbb{R}^{6 \times H \times W}$ comprises the spatially broadcasted Plücker ray embeddings for geometric conditioning.

This geometric conditioning proves critical when upstream generators fail to maintain left-right consistency, often producing visually similar or identical images for different viewpoints. The Plücker embeddings enable the model to disambiguate view relationships and prevent refinement failures that would otherwise occur due to insufficient geometric constraints.

**Dual-Pathway Attention Mechanism**   Our dual-pathway attention mechanism represents the core architectural innovation that enables simultaneous preservation of fine visual details through local alignment while enforcing global geometric coherence across the entire view set. This design addresses the inherent trade-off between visual fidelity and geometric consistency that constrains existing monolithic approaches.

**Local Geometry Alignment** enforces view-specific correspondence by establishing the front view as the primary geometric reference, motivated by the observation that text-to-image generators typically produce the most accurate representation in the initial front view.Liu et al. (2023); Zhang et al. (2025); Ahn et al. (2024) For each diffusion block output $\mathbf{f}_i$ where $i \in \{\text{front, left, back, right}\}$, we compute the query, key, and value projections:

$$\mathbf{Q}_i = \mathbf{f}_i \mathbf{W}_Q, \tag{2}$$
$$\mathbf{K}_{front} = \mathbf{f}_{front} \mathbf{W}_K, \tag{3}$$
$$\mathbf{V}_{front} = \mathbf{f}_{front} \mathbf{W}_V \tag{4}$$

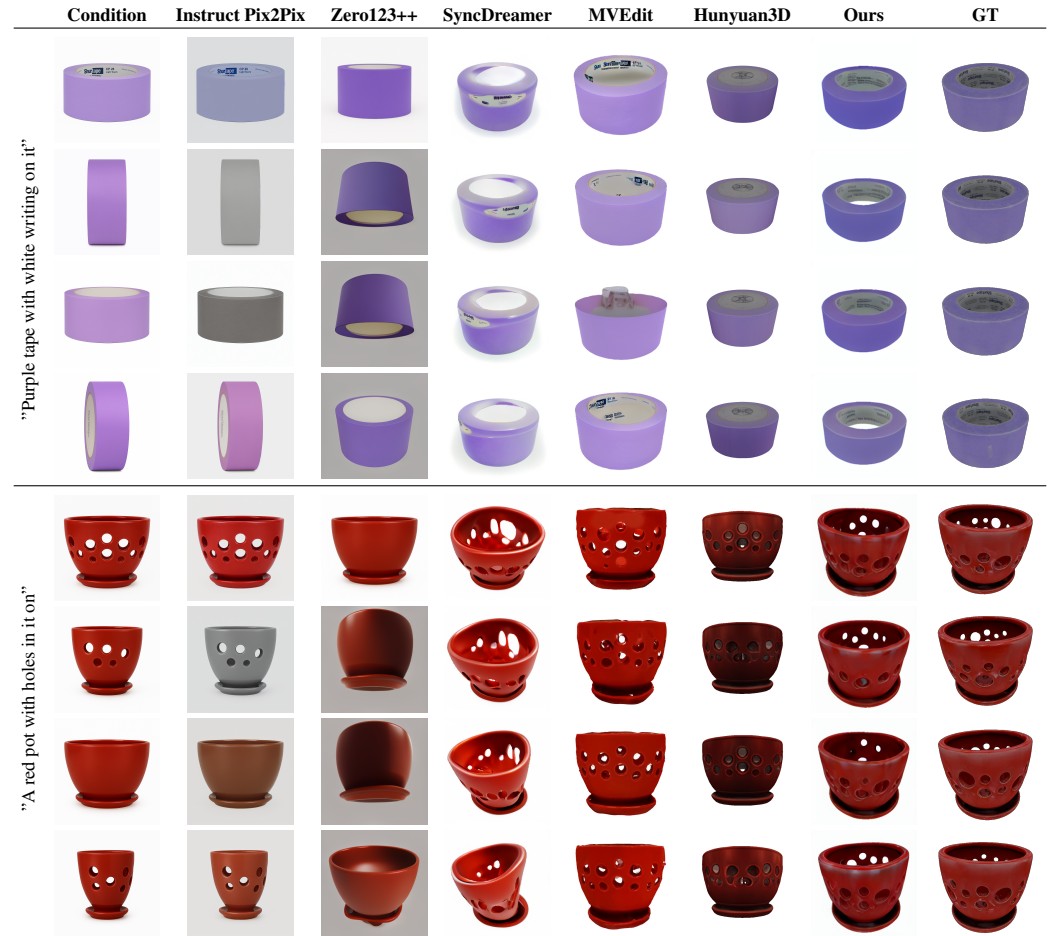

Figure 4: Multi-view comparison demonstrating MV-Diffus3R's geometric refinement capabilities against existing methods. Given identical GPT-generated input conditions, our approach achieves superior view consistency and detail preservation across four orthogonal viewpoints compared to current text-to-3D and image-to-3D techniques. GT column shows ground truth reference images.

The local alignment features are then obtained through cross-attention for non-front views:

$$\mathbf{f}_{local}^{(i)} = \text{CrossAttention}(\mathbf{Q}_i, \mathbf{K}_{front}, \mathbf{V}_{front}) \tag{5}$$

For the front view itself, we apply self-attention to maintain feature consistency:

$$\mathbf{f}_{local}^{(\text{front})} = \text{SelfAttention}(\mathbf{f}_{front}) \tag{6}$$

**Global Geometry Alignment** complements the local alignment and prevents over-dependence on the front view by providing each view with access to holistic geometric information. We compute a global feature representation by encoding all four input views through a VAE encoder and performing element-wise averaging:

$$\mathbf{g}_{global} = \frac{1}{4} \sum_{i=1}^{4} \text{VAE}_{\text{enc}}(\mathbf{I}_i) \tag{7}$$

where $\mathbf{I}_i$ represents the input image for the $i$-th view.

The global key-value projections are computed as:

$$\mathbf{K}_{global} = \mathbf{V}_{global} = \mathbf{g}_{global} \mathbf{W}_{global} \tag{8}$$

Each diffusion block output then performs cross-attention with this global feature:

$$\mathbf{f}_{global}^{(i)} = \text{CrossAttention}(\mathbf{Q}_i, \mathbf{K}_{global}, \mathbf{V}_{global}) \tag{9}$$

The final output feature for each view combines both pathways through residual connection:

$$\mathbf{f}_{output}^{(i)} = \mathbf{f}_{input}^{(i)} + \mathbf{f}_{local}^{(i)} + \mathbf{f}_{global}^{(i)} \tag{10}$$

This dual-pathway architecture ensures that each view benefits from both targeted front-view alignment and comprehensive global geometric context, enabling effective refinement while maintaining detail preservation. The modular design allows seamless integration with any existing text-to-image or single-view-to-multi-view generation system without requiring model modifications or retraining, establishing a framework that can evolve with advances in 2D generation technology.

## 4 EXPERIMENTS

We evaluate MV-Diffus3R through comprehensive quantitative and qualitative analysis, demonstrating superior geometric refinement capabilities while maintaining visual fidelity.

### 4.1 EXPERIMENTAL SETUP

**Implementation Details.** Training employed 640,000 steps with batch size 4 across 8 NVIDIA V100 GPUs over 5 days. The model initializes from InstructPix2Pix weights using standard diffusion $\epsilon$-prediction:

$$\mathcal{L} = \mathbb{E}_{t, \mathbf{x}_{GT}, \epsilon \sim \mathcal{N}(0, \mathbf{I})} \left[ \| \epsilon - \epsilon_\theta(\mathbf{x}_t, \mathbf{x}_{SV3D}, c_{text}, c_{ray}, t) \|^2 \right]$$

Inference establishes the front view as primary alignment reference with BLIP2-generated captions Li et al. (2023b). DDIM inversion generates 14-channel inputs (4 noise latents, 4 VAE-encoded conditioning views, 6 Plücker ray embeddings) requiring 16GB VRAM with FP16 precision.

**Evaluation Protocol.** We evaluate on Google Scanned Objects Downs et al. (2022) with GPT-generated multi-view sets (4,000 evaluation images). Baselines include Zero-1-to-3++, Sync-Dreamer, MVEdit Chen et al. (2024), Hunyuan3D 2.0 Zhao et al. (2025), and InstructPix2Pix fine-tuning. We assess performance using eight metrics: FID, CLIP, DINO Oquab et al. (2024), PSNR, SSIM, and LPIPS for image quality assessment; ULIP Xue et al. (2023) and Uni3D Zhou et al. (2023) for 3D reconstruction quality.

### 4.2 QUANTITATIVE RESULTS

| Method | FID ↓ | CLIP ↑ | DINO ↑ | PSNR ↑ | SSIM ↑ | LPIPS ↓ |
|---|---|---|---|---|---|---|
| Original GPT Image | 327.082 | 0.501 | 0.551 | 8.894 | 0.4906 | 0.5086 |
| Instruct Pix2Pix | 327.935 | 0.511 | 0.536 | 9.3337 | 0.4983 | 0.536 |
| Zero123++ | 355.711 | 0.540 | 0.484 | 9.4253 | 0.4772 | 0.5733 |
| SyncDreamer | 349.864 | 0.530 | 0.527 | 9.096 | 0.4798 | 0.5108 |
| MVEdit | 316.758 | 0.528 | 0.612 | 10.7757 | 0.5215 | 0.4574 |
| Hunyuan 3D 2.0* | 230.611 | 0.666 | 0.698 | 9.7887 | 0.504 | 0.4346 |
| Instruct Pix2Pix (Finetuned)† | 193.506 | 0.784 | 0.662 | 14.2127 | 0.6116 | 0.2259 |
| **Ours (MV-Diffus3R)** | **154.915** | **0.785** | **0.772** | **14.5717** | **0.6345** | **0.196** |

*Uses native reconstruction pipeline; all other methods use Trellis for mesh generation.
†Finetuned on distorted images generated by SV3D for domain adaptation.

Table 1: Quantitative comparison of multiview refinement methods on the GSO evaluation dataset. Image quality metrics (FID, CLIP, DINO, PSNR, SSIM, LPIPS) assess visual fidelity, semantic consistency, and perceptual quality of refined views. Lower FID and LPIPS scores and higher values for all other metrics indicate superior performance.

Table 1 demonstrates MV-Diffus3R's substantial improvements across all metrics. Our method achieves the lowest FID (154.915, 33% improvement over Hunyuan3D 2.0), highest CLIP (0.785) and DINO (0.772) scores, indicating superior semantic consistency and detail preservation. The new image quality metrics further validate our approach: PSNR (14.57), SSIM (0.635), and LPIPS

| Method | ULIP-I ↑ | Uni3D-I ↑ |
|---|---|---|
| Original GPT Image | 0.109 | 0.554 |
| Instruct Pix2Pix | 0.111 | 0.579 |
| Zero123++ | 0.117 | **0.605** |
| SyncDreamer | 0.119 | 0.603 |
| MVEdit | 0.114 | 0.580 |
| Hunyuan 3D 2.0* | 0.127 | 0.597 |
| Instruct Pix2Pix (Finetuned)[†] | 0.126 | 0.592 |
| **Ours (MV-Diffus3R)** | **0.128** | 0.597 |

*Uses native reconstruction pipeline.

[†]Finetuned on distorted images.

Table 2: 3D reconstruction quality comparison on the GSO evaluation dataset using geometric understanding metrics. All methods except Hunyuan 3D 2.0 use Trellis for mesh generation. Higher values indicate better geometric reconstruction quality.

(0.196) significantly outperform existing methods, demonstrating enhanced perceptual quality and reduced distortion. For 3D assessment, we achieve the highest ULIP-I score (0.128) and competitive Uni3D-I performance (0.597), confirming that geometrically refined images improve downstream reconstruction quality.

Traditional single-view-to-multi-view methods struggle with geometric consistency when provided with distorted inputs. While Hunyuan3D 2.0 shows reasonable performance, it cannot match our refinement quality. The InstructPix2Pix baseline demonstrates the necessity of specialized geometric conditioning, as naive fine-tuning fails to address multi-view consistency effectively.

## 4.3 ABLATION STUDIES

| Component Integration | | | | Image Quality | | | 3D Reconstruction Quality | |
|---|---|---|---|---|---|---|---|---|
| Inv. | Plücker | Local | Global | FID ↓ | CLIP ↑ | DINO ↑ | ULIP-I ↑ | Uni3D-I ↑ |
| | | | | 193.506 | 0.784 | 0.662 | 0.126 | 0.592 |
| ✓ | | | | 185.690 | 0.771 | 0.742 | 0.123 | 0.585 |
| ✓ | ✓ | | | 187.053 | 0.777 | 0.745 | 0.120 | 0.581 |
| ✓ | ✓ | ✓ | | 162.892 | 0.767 | 0.765 | 0.128 | **0.598** |
| ✓ | ✓ | ✓ | ✓ | **154.915** | **0.785** | **0.772** | **0.128** | 0.597 |

Table 3: Ablation study demonstrating the progressive contribution of each architectural component across image quality and 3D reconstruction metrics. Checkmarks indicate which modules are incorporated in each configuration. All model variants were trained for 640,000 steps for fair comparison.

We conduct systematic ablation using additive methodology, incrementally integrating proposed modules. Table 3 and Figure 5 present quantitative and visual analysis.

DDIM inversion yields immediate improvements across FID, DINO, PSNR, and SSIM, demonstrating enhanced detail preservation. Plücker ray embeddings provide crucial geometric disambiguation, evidenced by stabilized ULIP-I scores for visually similar views. The Local Alignment module shows significant improvements by establishing front-view geometric reference, with enhanced CLIP scores reflecting improved semantic consistency. The Global Alignment module prevents front-view overfitting through holistic geometric context, achieving optimal balance between consistency and detail preservation across all metrics including the newly added PSNR, SSIM, and LPIPS measures.

## 4.4 QUALITATIVE ANALYSIS

Figure 4 demonstrates superior geometric consistency and detail preservation compared to baselines. Traditional methods exhibit significant artifacts when processing distorted inputs, while our refine-

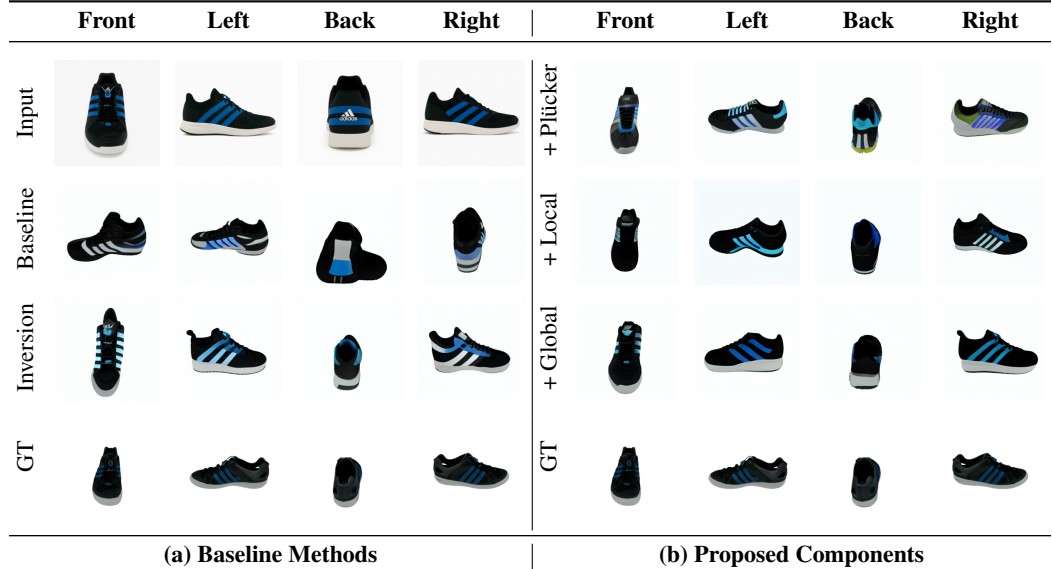

| | Front | Left | Back | Right | | Front | Left | Back | Right |

(a) Baseline Methods | (b) Proposed Components

Figure 5: **Ablation study comparing baseline methods against our proposed components.** The table presents a direct side-by-side comparison. **(a) Left:** Baseline and inversion methods fail to correct severe geometric inconsistencies across the four views. **(b) Right:** Our components progressively improve multi-view consistency. Adding Plücker Ray guidance, Local Alignment (+ Local), and Global Alignment (+ Global) systematically enhances the geometry, with our final result closely matching the Ground Truth (GT).

ment successfully corrects inconsistencies while maintaining high-frequency details. The ablation visualization illustrates progressive refinement through component integration, with each module contributing to improved geometric coherence without compromising visual quality.

### 4.5 LIMITATIONS

Our method faces constraints when upstream generators produce identical images across viewpoints—common with complex scenes or certain prompts. While Plücker embeddings provide disambiguation, lack of visual variation constrains meaningful geometric relationship inference. This limitation affects any multi-view refinement method, highlighting the importance of diverse initial view generation. Despite this constraint, our method demonstrates substantial improvements across the majority of evaluation cases.

## 5 CONCLUSION

We present MV-Diffus3R, a plug-and-play post-refinement module that addresses geometric inconsistencies in multi-view images generated by large-scale text-to-image models. Our two-stage pipeline decouples initial view generation from geometric refinement, enabling effective utilization of existing 2D foundation models without modification. The core contribution lies in specialized conditioning using Plücker ray embeddings and dual-pathway attention to enforce geometric coherence while preserving visual details. Experimental results demonstrate substantial improvements in handling characteristic distortions from diffusion-based generation systems, particularly addressing feature loss during rotational view synthesis. Our method achieves superior performance metrics across multiple evaluation benchmarks, showing significant enhancement in geometric consistency without sacrificing visual quality.The proposed approach represents a novel paradigm for 3D mesh generation workflows, suggesting that dedicated post-processing modules can effectively bridge powerful but geometrically inconsistent 2D generators with downstream 3D reconstruction systems.

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

## A  LARGE LANGUAGE MODEL USAGE DISCLOSURE

All experimental methodology, architectural innovations, and research contributions presented in this paper are entirely our original design and implementation. The core concept of optimizing GPT-generated multi-view images for 3D geometric consistency, including the MV-Diffus3R architecture with Plücker ray embeddings and dual-pathway attention mechanisms, represents our independent research contribution.

As non-native English speakers, we first composed the entire manuscript in English ourselves, including all scientific content and experimental design. We subsequently used Large Language Models solely to polish sentence structure and correct grammatical errors. The LLM assistance was limited to linguistic refinement and did not contribute to research methodology or scientific conclusions.

Our evaluation methodology intentionally uses GPT's image generation to create multi-view test sets with Google Scanned Objects (GSO) ground truth. This design choice aligns with our research objective of refining GPT-generated outputs for improved geometric consistency, ensuring practical relevance of our experimental validation.

## B  MULTI-VIEW GENERATION METHODOLOGY

This appendix provides comprehensive implementation details, dataset documentation, and additional experimental analysis to support the main paper. The content is organized into four sections that detail our multi-view generation methodology, technical implementation specifications, dataset construction process, and boundary case analysis.

### B.1  STRUCTURED PROMPTING STRATEGY

Our multi-view generation approach employs a hierarchical prompting strategy that separates global consistency constraints from view-specific requirements. This architectural decision enables superior geometric coherence across generated views while maintaining fine-grained object details.

#### B.1.1  PROMPT TEMPLATE ARCHITECTURE

The following structured template governs multi-view image generation with GPT's model:

> *Generate a 0° Front View of [User Input Description] using ChatGPT image generation tools*
>
> **Global Consistency Constraints:**
>
> - Maintain perfect object centering across all viewpoints
> - Preserve consistent scale throughout the view sequence
> - Eliminate perspective distortion, tilt, and skew artifacts
> - Apply uniform white background without shadows
> - Generate square 1024×1024 resolution outputs
> - Ensure surface detail, texture, and color consistency
>
> **View-Specific Parameters:**
>
> - Orient object directly facing the viewer (0° azimuth)
> - Establish canonical reference for subsequent view generation

This hierarchical specification enables the model to maintain global coherence while adapting to view-specific requirements, resulting in significantly improved multi-view consistency compared to unstructured approaches.

| Front | Left | Back | Right |

*Prompt: "An unscrambled Rubik's cube"*

Figure 6: Multi-view generation failure without structured prompting. The model produces inconsistent geometry, scale variations, and view-dependent artifacts when global constraints and view-specific instructions are absent.

### B.1.2 IMPACT OF STRUCTURED PROMPTING

Figure 6 demonstrates the critical importance of structured prompting for multi-view generation. Without explicit constraint separation, text-to-image models produce systematic failures including scale drift, orientation inconsistencies, and progressive detail loss that fundamentally compromise geometric integrity.

## C  IMPLEMENTATION SPECIFICATIONS

### C.1  TRAINING CONFIGURATION

Table 4 presents the complete hyperparameter configuration employed during model training. These parameters were optimized through systematic grid search on our validation dataset, balancing computational efficiency with model performance.

| Parameter | Value |
|---|---|
| Learning Rate | 1e-4 |
| Optimizer | Adam |
| Adam $\beta_1$ | 0.9 |
| Adam $\beta_2$ | 0.95 |
| Adam $\epsilon$ | 1e-06 |
| Adam Weight Decay | 1e-2 |
| Batch Size | 4 |
| Random Seed | 42 |
| Loss Function | L1 |
| Gradient Clipping | 10.0 |
| LR Scheduler | Cosine Annealing |
| Warmup Steps | 10,000 |
| Validation Split | 10% |
| Training Steps | 640,000 |
| Mixed Precision | FP16 |
| Gradient Checkpointing | Enabled |
| Memory Efficient Attention | XFormers |

Table 4: Training hyperparameter configuration for MV-Diffus3R. The batch size of 4 reflects memory constraints from the 14-channel input tensor incorporating Plücker ray embeddings.

## C.2 Model Architecture Details

Table 5 specifies the architectural parameters governing our geometric conditioning and attention mechanisms. These configurations were selected to optimize the balance between computational efficiency and geometric refinement quality.

| Component | Configuration |
| --- | --- |
| Attention Heads | 8 |
| Attention Dropout | 0.2 |
| Noise Prediction | $\epsilon$-parameterization |
| EMA | Enabled |
| Text Guidance Scale | 7.0 |
| Image Guidance Scale | 2.5 |
| Text Encoder | CLIP ViT-B/32 |
| 3D Reconstruction | TRELLIS-image-large |

Table 5: Architecture specifications for geometric conditioning and inference.

# D Dataset Documentation

## D.1 Training Dataset Construction

Our training dataset pairs SV3D-generated distortions with geometrically consistent ground truth renderings from Objaverse XL. This approach creates controlled geometric inconsistency patterns essential for learning effective refinement strategies. Figure 7 illustrates representative examples from our training dataset, demonstrating the systematic distortions that our method learns to correct.

The SV3D generation process introduces diverse geometric artifacts including asymmetric features, shape deformations, and cross-view detail inconsistencies that provide comprehensive training scenarios for our refinement model.

## D.2 Evaluation Dataset Characteristics

The evaluation dataset comprises GPT's generated multi-view sets paired with Google Scanned Objects ground truth, capturing real-world geometric inconsistencies encountered in production workflows. Figure 8 presents examples that demonstrate the characteristic artifacts arising from text-based view generation commands.

These evaluation examples represent practical deployment scenarios, exhibiting progressive detail degradation, left-right consistency failures, and rotational ambiguities inherent to text-based view generation systems.

# E Boundary Case Analysis

While MV-Diffus3R demonstrates robust performance across diverse object categories, we identify specific boundary cases that present challenges for geometric refinement. Figure 9 illustrates two primary scenarios where refinement effectiveness is reduced.

The first scenario involves multi-object scenes where spatial relationships between discrete entities cannot be properly disambiguated through our single-object optimization approach. The second scenario occurs when upstream generators produce visually similar views across different viewpoints, preventing the establishment of meaningful geometric correspondences despite Plücker ray conditioning.

These boundary cases inform practical deployment considerations and highlight the importance of appropriate input generation for optimal refinement results. When upstream generators provide sufficient visual variation and single-object focus, our method consistently achieves high-quality geometric refinement suitable for downstream 3D reconstruction applications.

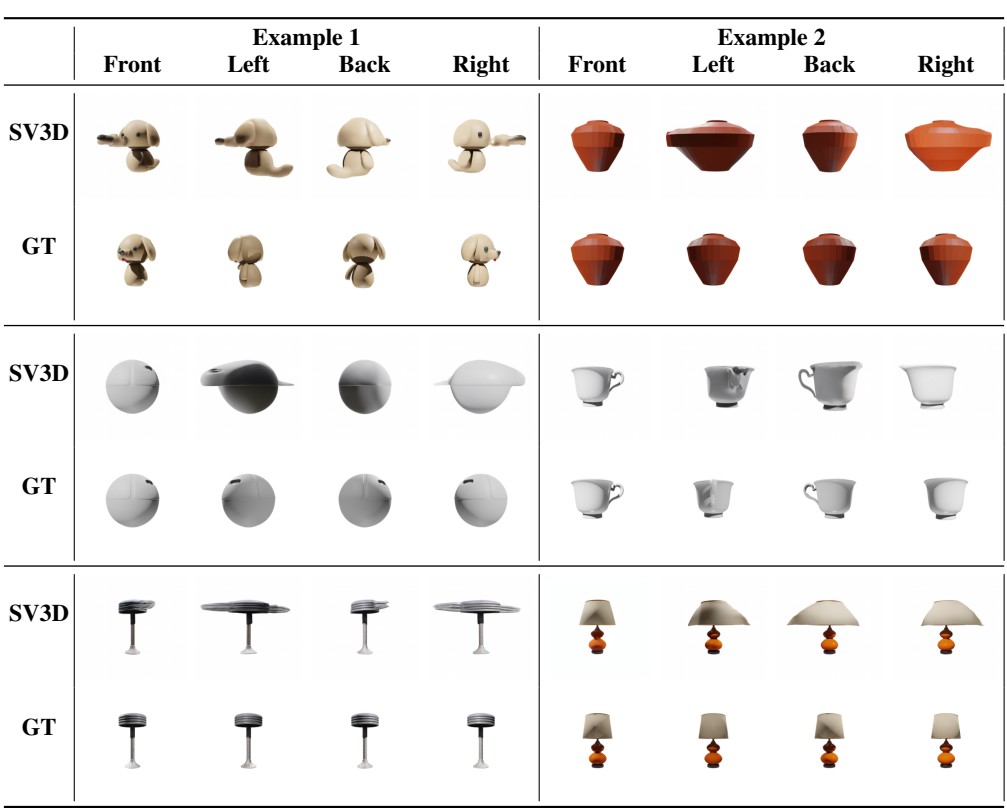

Figure 7: Training dataset examples showing SV3D-generated distortions (SV3D rows) paired with ground truth renderings from Objaverse XL (GT rows). Each example demonstrates characteristic geometric inconsistencies including asymmetric features, shape deformations, and cross-view detail loss that our method learns to correct.

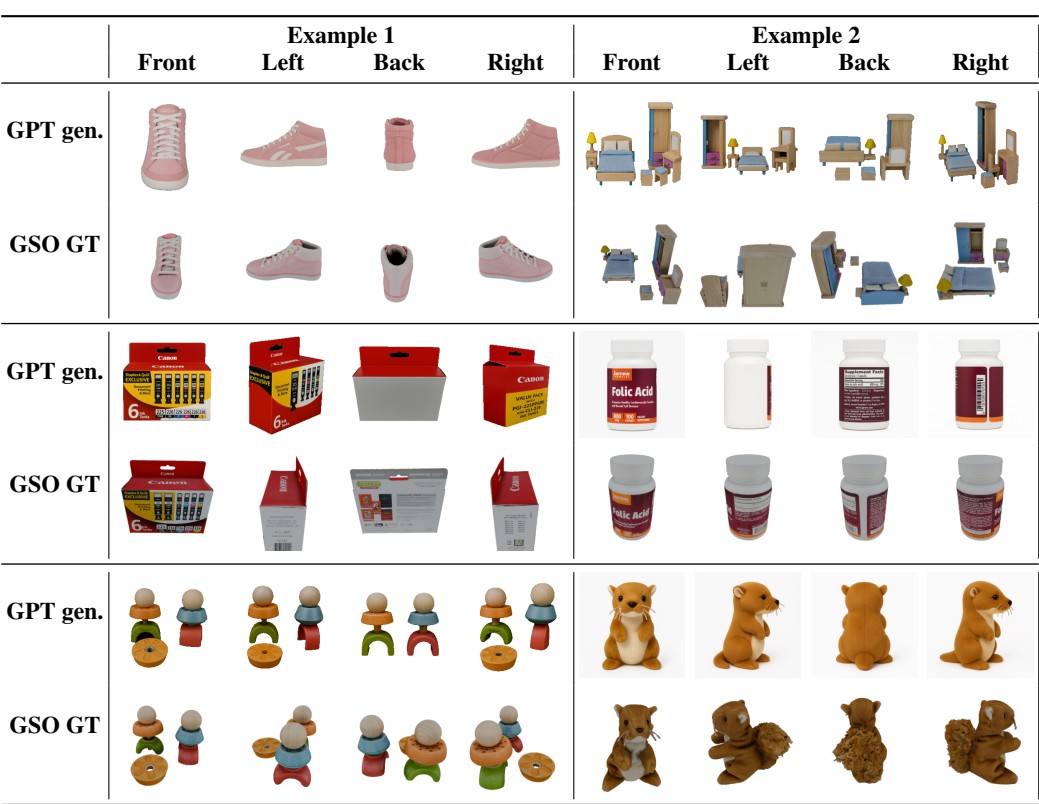

Figure 8: Evaluation dataset examples comparing GPT generated multi-view generation with Google Scanned Objects (GSO GT) ground truth. The GPT gen. rows demonstrate characteristic inconsistencies from text-based rotation commands, including detail loss across views, left-right confusion, and perspective shifts that impact 3D reconstruction quality.

| Example 1: Multi-object Scene | | | |
| --- | --- | --- | --- |
| | Front | Left | Back | Right |
| Input |  | | | |
| Output | | | | |
| Issue | Multiple objects confuse geometric correspondence | | | |

| Example 2: Visually Similar Views | | | |
| --- | --- | --- | --- |
| | Front | Left | Back | Right |
| Input |  | | | |
| Output | | | | |
| Issue | Insufficient visual variation prevents geometric disambiguation | | | |

| Example 3: Severe Occlusion | | | |
| --- | --- | --- | --- |
| | Front | Left | Back | Right |
| Input |  | | | |
| Output | | | | |
| Issue | Occlusion limits refinement capability | | | |

Figure 9: Limitation analysis showing three challenging scenarios for MV-Diffus3R. The method encounters reduced effectiveness when handling: (1) multi-object scenes where spatial relationships between entities cannot be properly disambiguated, (2) visually similar views where insufficient variation across viewpoints prevents the model from establishing meaningful geometric correspondences despite Plücker ray conditioning, and (3) severe occlusion with multiple objects that provide insufficient visual cues for geometric alignment. These cases represent fundamental boundaries of refinement-based approaches.

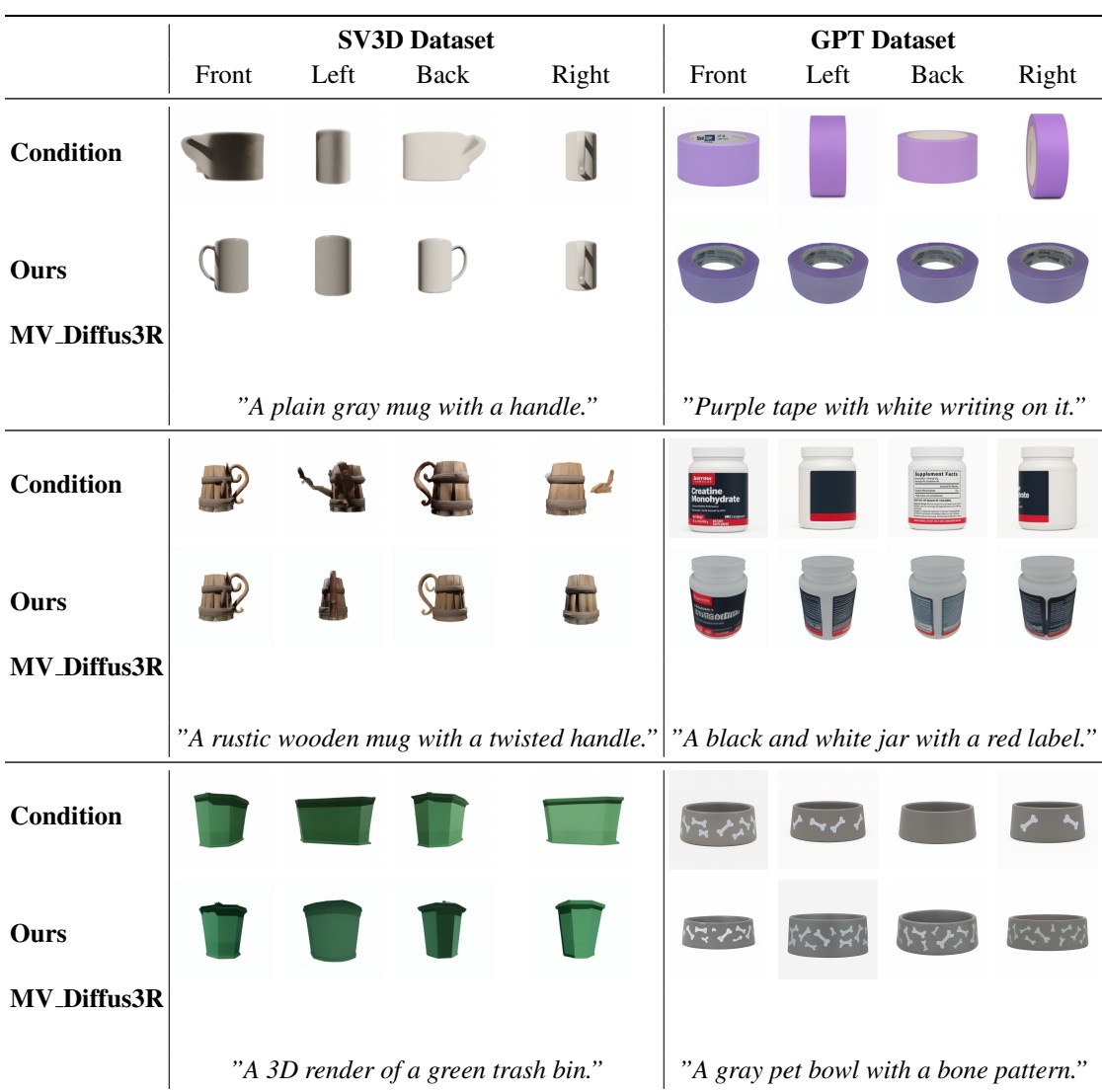

| | SV3D Dataset | | | | GPT Dataset | | | |
|---|---|---|---|---|---|---|---|---|
| | Front | Left | Back | Right | Front | Left | Back | Right |
| **Condition** | | | | | | | | |
| **Ours** | | | | | | | | |
| **MV_Diffus3R** | | | | | | | | |
| | *"A plain gray mug with a handle."* | | | | *"Purple tape with white writing on it."* | | | |
| **Condition** | | | | | | | | |
| **Ours** | | | | | | | | |
| **MV_Diffus3R** | | | | | | | | |
| | *"A rustic wooden mug with a twisted handle."* | | | | *"A black and white jar with a red label."* | | | |
| **Condition** | | | | | | | | |
| **Ours** | | | | | | | | |
| **MV_Diffus3R** | | | | | | | | |
| | *"A 3D render of a green trash bin."* | | | | *"A gray pet bowl with a bone pattern."* | | | |

Figure 10: **Multi-view refinement comparison across SV3D and GPT-generated datasets.** Our method demonstrates consistent geometric refinement capabilities across diverse input sources and object categories. **Left panels:** Results on SV3D-generated distorted views showing correction of characteristic single-view-to-multi-view artifacts including asymmetric features and shape deformations. **Right panels:** Results on GPT-generated multi-view sets demonstrating refinement of text-based rotation inconsistencies and left-right disambiguation failures. For each example, *Condition* rows show input multi-view sets with geometric distortions, while *Ours* rows present MV-Diffus3R refined outputs achieving improved cross-view consistency while preserving fine visual details. Corresponding 3D reconstruction results generated using the TRELLIS model are provided in the supplementary video materials, demonstrating improved mesh quality and geometric coherence achieved through our refinement approach.

