# OpenReview forum: "MV-Diffus3R: Refining Multi-View Diffusions for Geometric Coherence 3D Reconstruction"
_ICLR.cc/2026/Conference — Submitted to ICLR 2026_

### Official Review · Reviewer_7Mwq · 2025-10-19

**Soundness:** 2
**Presentation:** 3
**Contribution:** 2
**Rating:** 2
**Confidence:** 5

**Summary:**

This paper addresses the problem of geometric inconsistency in multi-view images generated by large-scale 2D text-to-image foundation models. The core of this pipeline is MV-Diffus3R, a novel plug-and-play refinement module which leverages Plücker ray embeddings for explicit geometric conditioning and a dual-pathway (local and global) attention mechanism to enforce cross-view coherence while preserving fine visual details. The method is trained to correct distortions from existing generators and is evaluated on multi-view sets generated by GPT. Experimental results demonstrate significant improvements in image quality and geometric consistency over existing methods, achieving 33% FID improvements while maintaining exceptional visual quality.

**Strengths:**

- The method is easy to follow, and the full training and architecture hyperparameter tables make it reproducible.
- Comprehensive empirical evaluations, encompassing both quantitative metric-based results and qualitative visual comparisons show MV-Diffus3R effectively improves multi-view geometric consistency.
- Large dual-dataset (~50k objects; ~1.05M images) combining controlled SV3D distortions with realistic GPT inconsistencies, strengthening statistical power and external validity.

**Weaknesses:**

- Although the method improves over the “Original GPT Image” (PSNR 8.894 → 14.5717), it omits a crucial baseline: single-view-to-multi-view models like SV3D, which are also used to synthesize the training data. A simpler yet stronger alternative is “high-quality single view → SV3D for the remaining views,” whose public metrics are often superior (e.g., PSNR=21.26, LPIPS=0.08, SSIM=0.88[1]). This raises the question: if direct generation already outperforms refinement of a weak multi-view set, why adopt the two-stage design? The paper should include direct quantitative comparisons against such baselines and articulate the unique advantages that MV-Diffus3R provides.
- The training distortions come from SV3D, but evaluation uses multi-view sets from GPT. As a specialized single-view-to-multi-view model, SV3D may induce geometric errors (e.g., shape deformations, asymmetry) that systematically differ from those of a general text-to-image model like GPT, creating a train–test mismatch.
- The paper's core technical contributions, namely the dual-pathway attention and the application of Plücker ray embeddings(Like EpiDiff[2]), demonstrate limited conceptual novelty. The work appears to be more of a skillful combination and application of established techniques, which may not fully meet the high expectations for fundamental innovation at a top-tier conference like ICLR.

[1]SV3D: Novel Multi-view Synthesis and 3D Generation from a Single Image using Latent Video Diffusion

[2]EpiDiff: Enhancing Multi-View Synthesis via Localized Epipolar-Constrained Diffusion

**Questions:**

- Why not pursue direct multi-view generation? If a single image to multiview images diffusion model (e.g., SV3D, MVDream) can produce superior multi-view sets from a single high-quality view, what is the motivation to train a separate refinement module instead?

**Details Of Ethics Concerns:**

If multi-view generation is the goal, why not directly use a dedicated single-image→multi-view model (e.g., SV3D, MVDream) instead of training a separate refinement module on outputs from a general text-to-image model? Please clarify the concrete advantages of the two-stage design.

---

### Official Review · Reviewer_JczP · 2025-10-29

**Soundness:** 2
**Presentation:** 3
**Contribution:** 2
**Rating:** 2
**Confidence:** 4

**Summary:**

This paper presents MV-Diffus3R, a plug-and-play module for refining geometrically inconsistent multi-view images from 2D foundation models. The method operates in a decoupled two-stage pipeline, leveraging an InstructPix2Pix backbone enhanced with plucker ray embeddings for geometric conditioning and a dual-pathway 3D-aware attention mechanism for cross-view alignment. The results demonstrate a better quality compared with direct multi-view generation, offering a post-hoc solution for multi-view generation pipelines.

**Strengths:**

The method is designed as a plug-and-play module that refines multi-view images without relying on camera pose estimation or 3D supervision. This design allows it to function as an independent post-processing step, which can be readily integrated into various multi-view generation pipelines. By operating under these conditions, the approach offers a practical and flexible solution for enhancing geometric consistency in existing workflows.

**Weaknesses:**

1. Motivation for a two-stage pipeline is not compelling. The core motivation of refining outputs from other multi-view models is not fully compelling. The primary challenge for models like MVDream is the trade-off between fidelity and coherence, largely stemming from limitations in 3D training data quality and scale. Since MV-Diffus3R is also trained on Objaverse, it does not fundamentally overcome this data bottleneck. This raises questions about the necessity of a separate refinement stage, as simply scaling up the base models' parameters and data might achieve similar improvements more efficiently.
2. Limited architectural novelty. The novelty of the proposed 3D-aware alignment module is not sufficiently established. Cross-view attention mechanisms for multi-view consistency, such as those in MVDream, are already prevalent. The paper lacks a thorough ablation or discussion that clearly delineates the distinct advantage of its specific dual-pathway (local vs. global) design over other existing cross-view interaction architectures.
3. Insufficient experimental validation. The qualitative results are limited to simple object categories and do not demonstrate performance on more complex scenes. Furthermore, the refined outputs exhibit a noticeable drop in brightness compared to the inputs, which is not addressed. This visual artifact and the lack of challenging cases make it difficult to fully assess the method's robustness and effectiveness.

**Questions:**

1. Could you provide more experimental analysis and insight into the dual-pathway attention design? For instance, a comparative study against other cross-view interaction mechanisms (e.g., a single unified attention) would help clarify the specific and distinct benefits contributed by the separate local and global alignment pathways.
2. Could you show more qualitative results on complex objects (e.g., with intricate geometry or multi-object scenes) to better demonstrate the model's refinement capability? Furthermore, the outputs appear to suffer from a noticeable reduction in brightness. Could you comment on the cause of this issue and potential solutions to mitigate it?

---

### Official Review · Reviewer_HQFF · 2025-10-30

**Soundness:** 3
**Presentation:** 3
**Contribution:** 2
**Rating:** 6
**Confidence:** 3

**Summary:**

The authors propose MV-Diffus3R, a framework designed to decouple multi-view image generation quality from geometric alignment performance.
To train the model, they utilize SV3D for handling misaligned multi-view image sets and employ ground-truth images from the Objaverse dataset.
The architecture incorporates Plücker Ray Embeddings, a Dual-Pathway Attention Mechanism, and a Local Geometry Alignment module to effectively model geometric consistency across views.
Experimental results demonstrate that the proposed method significantly improves multi-view image synthesis performance, achieving approximately 30% lower FID compared to existing methods such as InstructPix2Pix.

**Strengths:**

1. Clear Motivation
The motivation of the research is well-defined. The authors correctly identify that existing multi-view image synthesis models jointly handle the coupled tasks of image generation and geometric alignment, which can limit overall performance. Their approach to decouple these two stages provides a clear and logical direction, addressing a key bottleneck in current methods.

2. Comprehensive Ablation Study
The authors conduct a thorough ablation study on the proposed model architecture, effectively demonstrating the contribution of each component to the final performance. This detailed analysis helps validate the design choices and provides valuable insights into how individual modules influence the overall results.

**Weaknesses:**

1. Validation Limited to InstructPix2Pix
Although the proposed method is claimed to be a plug-and-play framework applicable to various image-to-multi-view synthesis models, the experiments are conducted only in combination with InstructPix2Pix. This narrow validation raises concerns about the generalization capability of the proposed approach. Additional experiments with other baseline models would strengthen the claim of model-agnostic applicability.

2. Dependence on SV3D for Training Data
The training pipeline relies heavily on SV3D to generate multi-view datasets. However, the paper does not include any ablation or comparative analysis using datasets generated by alternative multi-view synthesis models. Such an evaluation would be valuable to assess the robustness and data-dependence of the proposed method and to verify whether its performance improvements generalize beyond SV3D-based data.

**Questions:**

1. How the FID score changes with different multi-view image generation networks?

2. How much the inference time would be increased if we leverage MV-Diffus3R as re-alignment module?

---

### Official Review · Reviewer_eJvK · 2025-10-31

**Soundness:** 2
**Presentation:** 2
**Contribution:** 1
**Rating:** 4
**Confidence:** 4

**Summary:**

This paper proposes a post processing method to process some GPT generated multi-view images to better and geometry aligned images. The network structure is based on instructp2p.

**Strengths:**

1. The proposed method is a post processing method to convert GPT generated multi-view images to better multi-view images. Results show improvement based on FID and other metrics.

2. The method is easy to follow.

**Weaknesses:**

1. This paper does not feature many technical contributions. It simply trains an instruct-p2p model to do post-processing and applies some tricks, e.g., DDIM inversion of the input image.

2. In terms of paper format, I feel the images are too large and have a lot of white space. Also Table 4.3 is not well aligned.

3. The proposed method relies on the high-quality images generated by GPT. What if the backbone is changed to some open-source models like Qwen-Image? Using a closed-source model like GPT will make the experiments hard to reproduce.

**Questions:**

Please refer to the weaknesses.

---

### Meta-Review · Area_Chair_x1TX · 2025-12-28

**Summary:**

The work considers the problem of multi-view image generation using large scale 2d text-to-image models. The main challenge when using such models as the basis for any subsequent multi-view generation technique is ensuring that the generated views are geometrically and perceptually consistent. To address this challenge, the work proposes a two-stage pipeline in which the second stage module takes several geometrically inconsistent views as input and produces geometrically consistent variants.

**Reviewer Concerns:**

The reviewers raise numerous concerns around novelty, motivation of the chosen pipeline, use of SV3D for data gneration, and the generalizaiton of the method.

**Reviewer Scores:**

The authors have chosen not to engage in the rebuttal process, leaving the reviewers' concerns open.

---

### Decision · Program_Chairs · 2026-01-26

Reject